# Idiosyncratic evolvability among single-point ribosomal mutants towards multi-aminoglycoside resistance

Laura Sánchez-Maroto[1‡], Guillem A. Devin[1‡], Pablo Gella[1,2], Alejandro Couce[1*]

**1** Centro de Biotecnología y Genómica de Plantas (CBGP), Universidad Politécnica de Madrid (UPM), Madrid, Spain, **2** Facultad de Ciencias de la Salud, Universidad Rey Juan Carlos, Madrid, Spain

‡ These authors are co-first authors on this work.
* a.couce@upm.es

## Abstract

Newly-arising mutations can impact not only fitness but also an organism's capacity for further adaptation (*i.e.*, its evolvability). Understanding what determines evolvability differences is of great interest from both fundamental and applied perspectives. A general pattern observed across multiple microbes is that evolvability tends to decline with genotype fitness (*i.e.*, the "rule of declining adaptability"), typically attributed to epistatic rather than mutational differences among genotypes. Here, we investigate whether common *rpsL* point mutations in *Escherichia coli*, conferring streptomycin resistance, may potentiate or hinder adaptation towards secondary aminoglycosides. We find a version of the rule of declining adaptability in which initially more-fit genotypes experience higher effective beneficial mutation rates but smaller effect sizes than their less-fit counterparts. Genome sequencing reveals the ribosome and electron transport chain as primary targets for adaptation. Second-step mutations typically confer cross-resistance across aminoglycosides, and some even restore fitness costs in the absence of drugs. However, some genotypes deviate markedly from the overall pattern, being completely unable to develop resistance to the secondary aminoglycosides. Such idiosyncratic dead-ends, if common among other systems involving single-point mutants, would expand the pool of potential targets for strategies to promote evolutionary robustness in biotechnology and combat multidrug resistance in clinical microbiology.

### Author summary

Identifying features that predict an organism's ability to evolve (evolvability) is highly desirable in fundamental and applied microbial genetics. Here we show that outstanding disparities in evolvability can arise from single-point variants within a single gene. Specifically, we investigated whether common

**Data availability statement:** All data necessary to replicate the research findings is available at public data repositories (NCBI Sequence Read Archive, PRJNA1235639) and in the electronic supplementary material.

**Funding:** This work was supported by the Agencia Estatal de Investigación (Proyectos de I+D+i, PID2019-110992GA-I00 and PID2022-142857NB-I00 to AC.; Centros de Excelencia "Severo Ochoa", SEV-2016-0672 and CEX2020-000999-S to the host institute; FPI PhD studentship, PRE2018-086448 to LS), and a Comunidad de Madrid "Talento" Fellowship (2019-T1/BIO-12882, 2023-5A/BIO-28940 to AC). The funders had no role in study design, data collection and analysis, decision to publish, or preparation of the manuscript.

**Competing interests:** The authors have declared that no competing interests exist.

streptomycin-resistance mutations can facilitate or hinder adaptation against other aminoglycosides. Most mutations facilitated resistance evolution, prompting further resistance gains in proportion to initial resistance levels. However, a few cases totally diverged from this trend, acting as evolutionary dead-ends. These findings suggest a scenario in which strong, unpredictable genetic interactions stand out against a backdrop of predictable trends, bearing implications for ongoing debates on which model best captures global fitness patterns in microbes. From an applied standpoint, since streptomycin is commonly used in mycobacterial infections and agriculture, our results suggest that some circulating pathogenic strains may already be predisposed to developing multi-aminoglycoside resistance, potentially informing surveillance and intervention strategies.

## 1. Introduction

The capacity to undergo adaptive evolution, termed evolvability, is fundamental to biological entities, and much effort has focused on understanding its molecular determinants and the extent to which these can be shaped by natural selection [1–5]. Experiments with microbes reveal that even closely-related genotypes can exhibit substantial evolvability differences, typically reported as reproducibly distinct fitness gains over comparable adaptation periods in the same environment [6–10]. At least two broad sets of causes may underlie these disparities in evolvability. On the one hand, evolvability differences can just reflect differences in the appearance rate of new variation. These include both genome-wide and localized heterogeneity in mutation rates and spectra [11–14], variations in the mutational target size for altering a given trait [15–17], and the differential accessibility to amino acids via point mutations that the genetic code imposes [18], even among synonymous codons [19]. On the other hand, evolvability differences can emerge because the fitness effects of new mutations often vary depending on the specific genetic context in which they occur. This context-dependency (*i.e.*, epistasis) is understood as a consequence of the non-linear structural, metabolic, and regulatory interactions of new mutations with previous mutations [20–24] and genes [25–29].

Despite the many mechanisms potentially contributing to variations in evolvability, a consistent trend emerges across a wide variety of microbial systems: low-fitness genotypes tend to be more evolvable than their high-fitness counterparts, a phenomenon known as the rule of declining adaptability [30]. This trend is commonly attributed to the preponderance of antagonistic interactions among beneficial mutations, such that their benefits are smaller when combined rather than individually (*i.e.*, antagonistic [31], negative [32] or "diminishing returns" [33] epistasis). Multiple reasons can explain why antagonistic epistasis should be this widespread. A basic intuition is that there are physio-chemical limits to how much a trait can be improved, which naturally manifests as plateaus in phenotype-fitness maps. For instance, the benefit from increasing the activity of a single enzyme can rapidly saturate due to the rise of secondary costs (*e.g.*, instability [34], resource depletion [35]) or as a result

of the buffering response of other enzymes in its metabolic pathway [22,36,37]. However, antagonistic epistasis is also frequently observed in cases lacking a clear biophysical explanation [9,38,39]. One view interprets such instances as reflective of the modular organization of cellular traits, suggesting that beneficial mutations in seemingly unrelated loci can actually be improving the same biological process, whose functionality cannot be improved indefinitely [40,41]. A similar argument is that, at the extreme, fitness could be seen as a global trait, so mutations could exhibit antagonistic interactions solely mediated by the absolute fitness of the organism [38,42].

A completely different explanation for the widespread observation of antagonistic epistasis has gained traction in recent years. Several authors have realized that global patterns of antagonistic epistasis can emerge as a statistical inevitability from randomly distributed, idiosyncratic epistasis among individual mutations [43–45]. This property is a manifestation of the well-known statistical principle of regression to the mean: if new beneficial mutations have completely random effects across backgrounds, then high-fitness genotypes will, on average, experience mutations with smaller relative contributions to their already elevated fitness levels compared to low-fitness genotypes. The hypothesis of idiosyncratic epistasis, besides being more parsimonious, is capable of explaining a broader range of empirical observations [43]. Additionally, this view lends credence to the prospect of identifying specific genotypes that, due to outsized idiosyncratic epistasis, deviate strongly from the rule of declining evolvability. Such outliers, particularly those hindering further evolution in a given direction, represent a prospect of notable interest from an applied standpoint, as for instance in the search for new approaches to prevent unwanted evolution in clinical and industrial settings [46–49].

In this line, a few studies have begun to search for genotypic and phenotypic features in bacteria that may predict a reduced capability for evolving antibiotic resistance [48,50–53]; an endeavor that holds significant promise for informing surveillance and compound development targets [48,54]. An influential study adapted hundreds of single-gene knockouts in *E. coli* towards major classes of antibiotics, revealing a general trend for initially more sensitive genotypes to experience greater resistance gains [50]. This result indicates that the rule of declining adaptability observed for fitness can apply to a subsidiary trait such as antibiotic susceptibility — a major component of fitness, although not necessarily with a one-to-one correspondence. Of note, this study also identified genetic variants with marked deviations from the general trend, including variants that facilitated resistance evolution and, most importantly, variants that slowed down and almost suppressed resistance evolution. Such idiosyncratic evolvability variants have also been observed in other studies with antibiotics, even if a consistent trend towards declining adaptability was unclear [51–53]. Reasons for the failure to detect this trend include combining small sample sizes with high levels of divergence among strains (*i.e.*, long-term experimental evolution lines [51], natural isolates [53]), or the use of evolution protocols with uneven selective pressures [52].

To help illuminate how common declining adaptability patterns and evolvability suppressors are in antibiotic resistance evolution, we sought to set up a model system in which differences in evolvability could be expected among strains harboring single-point mutations, and in which evolvability differences could be tested along several relevant phenotypic directions. To this end, we turned to the relatively well-understood model system of spontaneous streptomycin resistance in *E. coli*. Streptomycin (STR) was the first described aminoglycoside, an important antibiotic class active against aerobic Gram-negative and some Gram-positive pathogens, most notably mycobacteria [55]. Aminoglycosides exert their bactericidal effect by binding to the 16S rRNA in the decoding region of the 30S small ribosomal subunit (the A-site) [56,57]. Although different compounds bind to different nucleotides on the 16S rRNA, aminoglycosides typically induce tolerance to codon-anticodon mismatches, leading to the production of faulty polypeptides that damage the membrane and ultimately cause cell death [58,59]. High-level STR resistance is readily attained through point mutations in a handful of conserved positions along *rpsL*, the gene encoding ribosomal protein S12, which interacts with the 16S rRNA at the decoding site [60,61]. These mutations often lead to the production of hyper-accurate yet slow ribosomes, imposing a substantial fitness cost in the absence of the drug [62,63].

While STR is still used in specific contexts, newer compounds such as the 2-deoxystreptamine (2-DOS) aminoglycosides are preferred due to their improved toxicity profiles and, most importantly, their extremely low rates of spontaneous

resistance mutation [64–66]. These low rates are explained by the fact that 2-DOS aminoglycosides bind not only to the 16S rRNA of the 30S small ribosomal subunit but also to the 23S rRNA of the 50S large ribosomal subunit [67,68]. As a consequence, point mutations in *rpsL* do not suffice to confer high-level 2-DOS aminoglycoside resistance, although some cause minor susceptibility changes in *E. coli* and *Mycobacterium smegmatis* [66,69]. In view of these considerations, we reasoned that some streptomycin-resistant *rpsL* mutations may act as stepping-stones toward 2-DOS aminoglycoside resistance, perhaps even requiring only one additional mutational step to achieve high-level resistance. Similar stepping-stone mechanisms have been described for other antibiotics for which resistance is difficult to attain via single-step mutations, such as methicillin in *Staphylococcus aureus* [70] and gepotidacin in enterobacteria [71].

Here, we challenged *rpsL* streptomycin-resistant mutants of *E. coli* with lethal concentrations of four different 2-DOS aminoglycosides. We found that single point mutations in *rpsL* can indeed serve as stepping-stones toward resistance against these compounds. However, the contribution to evolvability varied markedly across the different mutations. Overall, we observed that strains that are initially more susceptible tend to undergo lower mutation rates but larger resistance gains than their less susceptible counterparts, conforming to the rule of declining adaptability. Of note, a few mutations significantly diverged from these patterns, reducing evolvability to the point of impeding genotypes to sustain resistance mutations. We also characterized the phenotypic consequences and genetic basis of the observed 2-DOS aminoglycoside resistance.

## 2. Results

### (a) First-step mutants show marked differences in evolvability towards 2-DOS aminoglycoside resistance

To examine how single-point mutations in the same gene can alter evolvability along several relevant phenotypic directions, we isolated spontaneous streptomycin-resistant mutants in *E. coli* and verified through Sanger sequencing the status of their *rpsL* gene (Materials and Methods). We identified 9 unique alleles bearing mutations in 4 different sites, all mapping to the vicinity of the interface between protein S12 and the 16S rRNA (S1A Fig). Of these, 4 alleles displayed the extreme phenotype known as streptomycin dependence, whereby cells need to maintain the drug permanently bound to their ribosomes for growth [60,72]. We decided to discard these streptomycin-dependent alleles because the required presence of streptomycin may elicit uncontrolled interactions with the different aminoglycosides to be tested, as it has been reported previously [73]. The final set of genotypes comprised five different streptomycin-resistance mutations (amino acid substitutions K43N, K43T, K88E, K88R and P91Q, numbering from the start codon; S1B Fig), known from the literature to display a range of fitness effects in the absence of the drug [62,74,75]. We then used Minimal Inhibitory Concentration (MIC) assays to characterize the effects of these mutations on susceptibility to four different 2-DOS aminoglycosides (Materials and Methods). The compounds, chosen due to their clinical relevance, were amikacin (AMK), gentamicin (GEN), kanamycin (KAN) and tobramycin (TOB). We observed only minor differences in MIC values. When present, these differences were always in the direction of making cells less susceptible to the different drugs, including several instances of a 2-fold and even a 4-fold reduction compared to the ancestral strain.

To test whether the set of first-step mutants can serve as stepping-stones toward 2-DOS aminoglycoside resistance, we grew overnight cultures of the isolates and tested their ability to form colonies on solid medium supplemented with lethal concentrations of the different drugs (Fig 1A). We aimed for drug concentrations surpassing the MIC for all strains, yet low enough to facilitate the emergence of resistance mutations, ideally only among first-step mutants. To this end, we first determined the Mutant Prevention Concentration (MPC) for each compound [76], defined as the concentration that inhibits the occurrence of resistant mutants in the ancestral strain (Fig 1B). We observed marked variations in the MPCs across compounds: a mere 4- or 6-fold increase over the MIC was sufficient to inhibit the ancestor's capacity to produce resistant mutants for KAN, TOB, and AMK respectively; whereas up to a 12-fold increase was required for GEN. We then confirmed that, under these concentrations, many of the first-step mutants were capable of producing secondary resistant

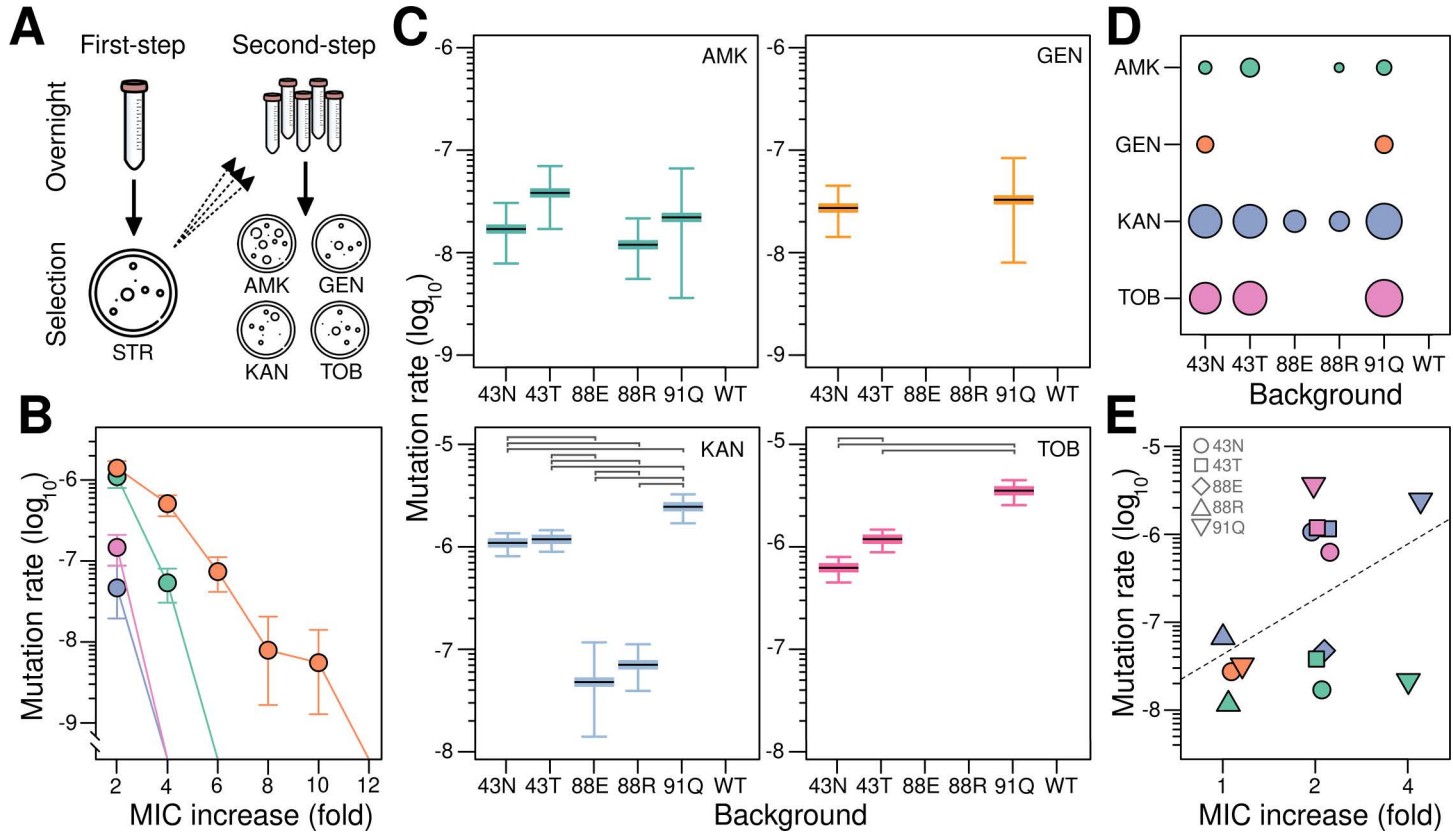

**Fig 1. Evolvability differences among *rpsL* mutants towards 2-DOS aminoglycoside resistance.** (A) Schematic overview of our two-step approach: first, selection of streptomycin-resistant mutants; second, selection of mutants additionally resistant to four different 2-DOS aminoglycosides. (B) Mutation rate for 2-DOS aminoglycoside resistance in the ancestral strain across a gradient of lethal concentrations of four compounds: amikacin (AMK, green), gentamicin (GEN, orange), kanamycin (KAN, blue), and tobramycin (TOB, pink). Antibiotic susceptibilities are presented as fold increases with respect to the Minimal Inhibitory Concentration (MIC) for the ancestral strain. Values represent the mutation rate ± 95% confidence interval, estimated from three parallel replicate cultures. (C) Mutation rate for 2-DOS aminoglycoside resistance in the first-step, streptomycin-resistant mutants. Values represent the mutation rate ± 95% confidence interval, estimated from five parallel replicate cultures. Selection was performed at the wild-type's Mutant Prevention Concentration (MPC), as informed by Panel B (AMK: 6×, GEN: 12×, KAN: 4×, TOB: 4×). Color and abbreviation convention as in (B). Horizontal bars denote statistically significant differences. Note that panels are arranged in upper and lower rows based on shared value ranges, but all span exactly three orders of magnitude to facilitate comparison; see (E) for direct comparison of values in a single plot. (D) Summary plot of the number of antibiotics for which a given streptomycin-resistance mutation can act as a stepping-stone towards 2-DOS aminoglycoside resistance. Circle areas are proportional to the mutation rates shown in (C). Color and abbreviation convention as in (B). (E) Relationship between initial susceptibility and mutation rates across antibiotics for the combinations of genotype and treatment that produced second-step mutants. Antibiotic susceptibilities are presented as fold increases with respect to the MIC measured in each background. The dashed line shows the best fit to a linear regression model. Color and abbreviation convention as in (B).

mutants to the different compounds (Fig 1C and 1D). Moreover, mutants showed substantial variation in their ability to develop the secondary resistances, with mutation rates ranging from several- to tens-fold differences across antibiotics. Notably, in 5 out of 20 cases mutants were completely incapable of producing any resistant mutant, thus representing evolutionary dead-ends rather than stepping-stones, at least within the resolution limits of our experiments (Materials and Methods).

To gain insight into whether these cases represent true dead-ends across a broader range of concentrations, we measured the MPC for the most extreme case (K88E). S2 Fig shows that MPC values are nearly identical to the wild-type for all antibiotics, except for kanamycin, where it increased slightly from 4 µg/mL to 6 µg/mL. This suggests that these

dead-ends result from a failure to enhance evolvability beyond wild-type levels rather than from a direct antagonistic interaction with secondary mutations. Note we use the term "dead-end" to describe genotypes from which no further beneficial mutations are available, thus terminating the line of adaptive evolution. Crucially, this term is defined within the context of our selective regime (i.e., lethal streptomycin followed by secondary aminoglycosides) and is therefore used in relation to alternative adapting variants—not the ancestor, which is eliminated at the outset. This usage is consistent with prior literature, which applies the term to the extreme case of genotypes that cannot adapt any further, while other alternative genotypes remain evolvable under the same selective conditions [5,50,77,78].

We next looked at what may be driving these marked differences in mutation rate. We first confirmed that these were not due to genome-wide elevations in mutation rate (S3 Fig), suggesting that differences stemmed from varying accessibility to beneficial mutations (i.e., target size). A simple explanation could be that target size correlates with the initial level of resistance: low-susceptibility strains are already closer to the necessary resistance threshold for growth, and therefore many minor-effect mutations may suffice to them to reach this threshold. Alternatively, target size could be idiosyncratic with respect to initial susceptibility, perhaps due to the specific structural and physicochemical details of how each mutation interacts with drugs and ribosomal components. We first examined whether initial susceptibility levels can explain the most notable observation; that some genotypes act as dead-ends rather than stepping-stones for further evolution. However, we found no significant association between initial susceptibility level and these two categories, defined by the presence or absence of resistant mutants across treatments ($P = 0.545$, Fisher's Exact Test). Next, we focused on the majority of cases (14/20) that did produce resistant mutants. Our analysis revealed a weak correlation between initial susceptibility and mutation rates across antibiotics (Pearson's $R = 0.44$; $P = 0.118$). In other words, a reduced resistance hurdle only partially explains why less-susceptible genotypes are more evolvable than more-susceptible ones, suggesting an important role for idiosyncratic epistatic interactions between genetic backgrounds and secondary mutations.

### (b) Second-step mutants show widespread cross-resistance to multiple aminoglycosides

We next examined the extent to which second-step mutations elicited changes in resistance beyond the antibiotic used for selection. In particular, we asked whether the second-step mutants exhibited common phenotypic traits across genotypes or, on the contrary, whether they displayed unique characteristics that may indicate specific physiological adaptations. To address this question, we selected three independent isolates from the 14 combinations of genotype and treatment that produced second-step mutants (Fig 1C and 1D) and assessed their potential trade-offs and trade-ups in susceptibility against a panel of relevant compounds. A first set of these compounds comprised aminoglycosides, including five 2-DOS aminoglycosides (the four used for selection plus neomycin, which belongs to a different subclass), and two non-2-DOS aminoglycosides (streptomycin and spectinomycin) [55]. In addition, the panel included two ribosome-targeting drugs from different families (chloramphenicol and tetracycline), and two compounds previously linked to altered aminoglycoside susceptibility (ciprofloxacin [79] and triclosan [80]).

Focusing on the aminoglycosides set, we observed different trends depending on the subclass. For 2-DOS aminoglycosides, there was a general tendency towards substantial cross-resistance (Fig 2), irrespective of the genotype or the drug used for second-step selection. While not truly an exception, a minor deviation from this pattern was observed in KAN-selected strains, which exhibited a lesser increase in resistance to AMK and TOB compared to the other treatments (*P<= 0.011* and *P<= 0.016*, respectively, pairwise Wilcoxon's rank-sum test, Benjamini-Hochberg corrected). As for the non-2-DOS aminoglycosides, two points merit consideration. First, the elevated streptomycin resistance of the first-step mutants tends to persist among the second-step mutants, except for strains carrying the P91Q mutation, which exhibit a hundred-fold decrease in streptomycin resistance after selection in KAN, TOB, and partially in GEN (Fig 2). Second, the trade-off between streptomycin and spectinomycin displayed by first-step mutants also persists in the second-step mutants, with only a handful of isolates showing 2- and 4-fold reductions in susceptibility (Fig 2).

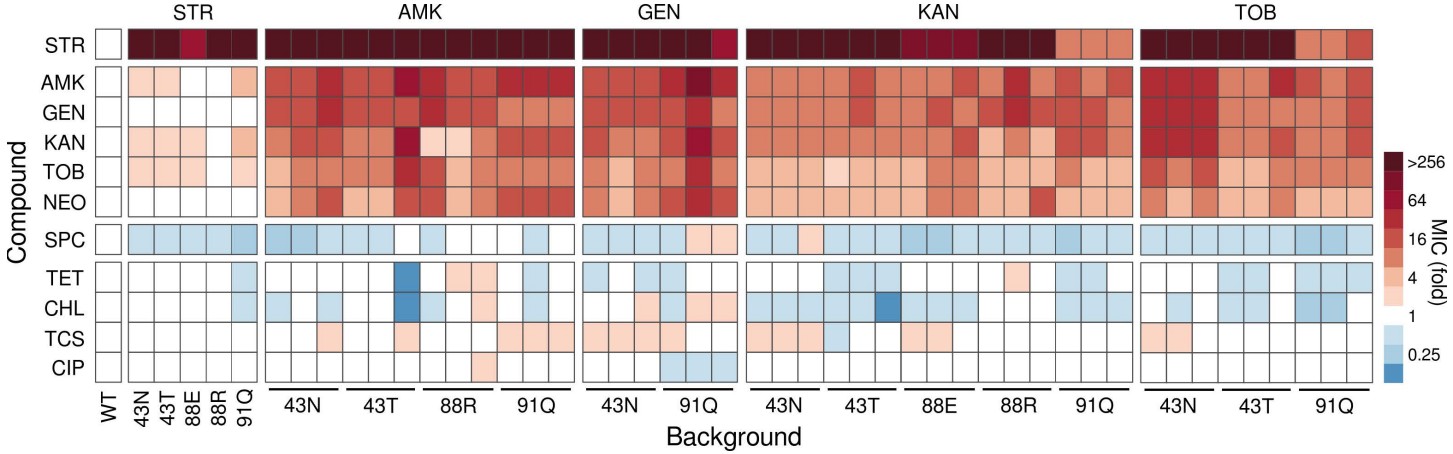

**Fig 2. Susceptibility profiles against multiple compounds for all strains studied in this work.** The heatmap shows Minimal Inhibitory Concentration (MIC) values for the collection of 48 strains comprising the ancestor, first-step and second-step mutants, across a panel of relevant antimicrobial compounds. These compounds are grouped by class, from top to bottom, starting with aminoglycosides (streptomycin, STR; amikacin, AMK; gentamicin, GEN; kanamycin, KAN; tobramycin, TOB; neomycin, NEO; and spectinomycin, SPC) and ending with non-aminoglycoside compounds (tetracycline, TET; chloramphenicol, CHL; triclosan, TCS; and ciprofloxacin, CIP). Strains are arranged by background, from left to right, starting with the ancestor, first-step mutants resistant to STR, and second-step mutants resistant to AMK, GEN, KAN, and TOB. Antibiotic susceptibilities are presented as fold increases relative to the MIC measured for the ancestor. White indicates no change compared to the ancestor, red indicates increased resistance, and blue indicates increased susceptibility.

Regarding the other compounds, trade-offs and trade-ups were much less common, and when they did occur, their magnitude was also substantially smaller, ranging from 4-fold increases to 2-fold reductions in susceptibility. Among these, the compounds for which we observed the clearest trend were the ribosome-targeting drugs chloramphenicol and tetracycline, with strains commonly becoming more susceptible after second-step selection (54.8% and 35.7% of cases, respectively; Fig 2). In contrast, cross-resistance was observed with triclosan in 38.1% of the isolates, and ciprofloxacin showed almost no changes across isolates (4/42, < 10%). Taken together with the results for the aminoglycoside set, these patterns suggest that genotypes may be acquiring similar physiological adaptations, possibly via mutations in the same genetic targets, or at least targeting the same cellular functions.

### (c) Second-step mutants show larger resistance gains in proportion to initial susceptibility levels

Next, we sought to use the above data to test whether the effect size of second-step mutations correlated with initial susceptibility to the aminoglycosides used for selection; in other words, whether or not they conform to the rule of declining adaptability. We observed that, indeed, strains tended to increase their MIC values in direct proportion to their initial level of susceptibility (Fig 3). Specifically, the most susceptible strains increased their MIC values by approximately eight times on average, twice as much of what was observed in the least susceptible strains, which showed only a 4-fold increase. Despite the discrete and relatively noisy nature of MIC values, this overall trend is robust: the correlation between initial and final susceptibilities was of moderate strength and statistical significance (Pearson's R = 0.5, P < 0.001) (Fig 3A). Moreover, when we normalized the data by the initial susceptibility level, focusing on the relative rather than absolute increase in effect size, we found a similar moderate correlation (Pearson's R = 0.37), which was still statistically significant (P = 0.016) (Fig 3B).

Taken together with the analyses of correlation with mutation rates, our data conforms to a version of the rule of declining adaptability in which initially less-susceptible genotypes experience higher effective beneficial mutation rates but smaller effect sizes than their more-susceptible counterparts. We note that in the context of studying declining adaptability

PLOS Genetics

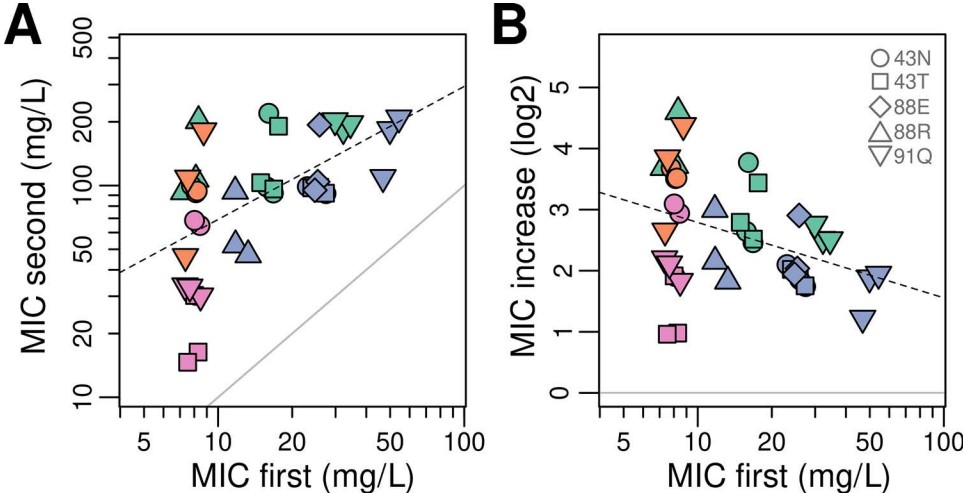

**Fig 3. The rule of declining adaptability among second-step mutants.** (A) Relationship between initial and final susceptibility to the 2-DOS amino-glycoside used for selection in each of the 14 genotype-treatment combinations that produced second-step mutants. Colors indicate the specific 2-DOS aminoglycoside used (AMK, green; GEN, orange; KAN, blue; TOB, pink), following the convention of Fig 1. The dashed line represents the best fit to a linear regression model. The solid gray line indicates the one-to-one correspondence; its slope serves as a reference for strains achieving the same increments irrespective of initial resistance. (B) The same data presented with final susceptibility as a relative fold increase, obtained by dividing each value by the corresponding initial susceptibility. The dashed line represents the best fit to a linear regression model. The solid gray line is the same as in panel (A); its slope here marks the independence of effect size from initial resistance. Note that a 10% uniform jitter was added to data points in both panels to aid visualization.

and evolvability suppressors, generating mutants via initial antibiotic exposure is arguably more realistic than engineering genome-wide knockouts [50,52]—suggesting that future research on similar model systems could enrich our understanding of antibiotic resistance evolution.

### (d) Second-step mutations can both mitigate and aggravate fitness costs in the absence of drugs

Given the ease with which common *rpsL* mutations can facilitate the acquisition of multi-aminoglycoside resistance, it is particularly important to evaluate the potential fitness costs associated with this phenotype. Fitness costs, observed in a variety of antibiotic resistance model systems, are of significant interest as they are expected to drive selection against resistance in antibiotic-free conditions, a property that may be leveraged on in the design of drug treatment protocols [81]. The single-point *rpsL* mutations used here have been a classic model in the study of the fitness cost of antibiotic resistance in different species [62,75,82]. These mutations typically show considerable variation in the cost of resistance, which can be substantial, a natural consequence of altering a key cellular component such as the ribosome. Therefore, we next sought to examine whether the second-step mutations exacerbate the fitness costs imposed by the original streptomycin resistance or, conversely, whether they have minor effects or can even mitigate fitness.

We conducted growth curve assays for all strains in the absence of antibiotics (S4 Fig) and used maximum growth rates as a proxy for fitness. Fig 4 confirms that first-step mutations typically incur substantial fitness costs, with the exception of K88R. By contrast, second-step mutations show a more variable range of fitness effects. Notably, only 40.5% of cases exhibit significant additional costs, while 35.7% show no significant change and 23.8% display fitness benefits (pairwise t-test, Benjamini-Hochberg corrected, assuming unequal variances). We found no clear correlation between the fitness of first-step mutants and the tendency of second-step mutations to have mitigating, neutral, or aggravating effects

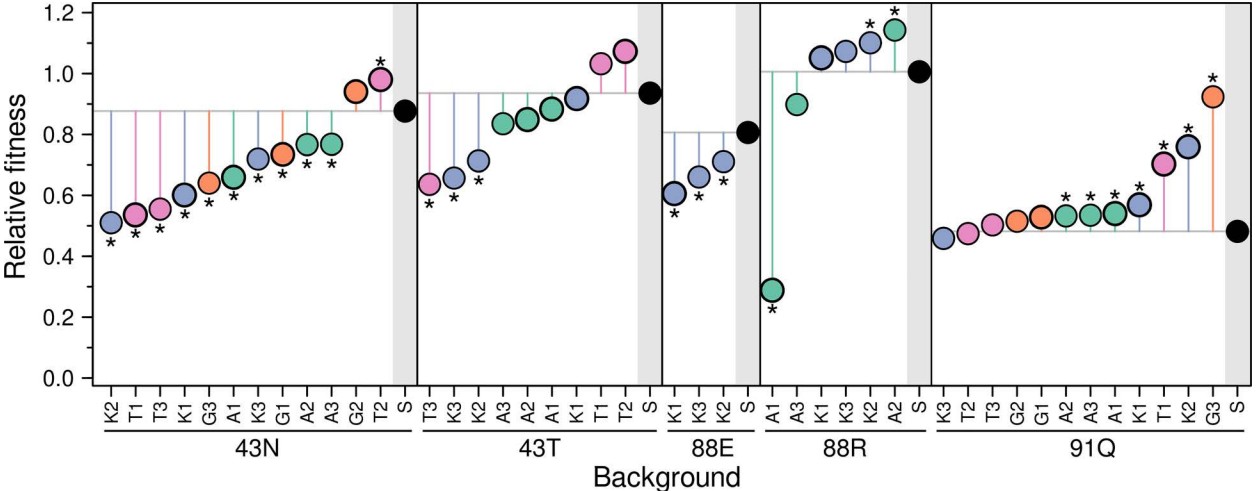

**Fig 4. Fitness effects of 2-DOS aminoglycoside resistance.** Points represent the fitness of each mutant relative to the ancestor, estimated as the ratio of their maximum growth rates in the absence of antibiotics (S4 Fig). Mutants are grouped into panels according to the identity of the first-step mutation. Within each panel, first-step mutants are highlighted with a grey shaded area, and their fitness set a reference line that helps visualizing whether second-step mutations mitigate or aggravate fitness costs. Second-step mutant labels consist of the first letter of the antibiotic used for selection and a number indicating the isolate. Colors represent the different antibiotics (AMK, green; GEN, orange; KAN, blue; TOB, pink; STR, black), following the convention of Fig 1. Asterisks indicate statistically significant differences. Values represent the mean of three replicates. Standard deviations are small and omitted for clarity but are available in the supplementary dataset. Strains subjected to whole-genome sequencing are indicated with a thicker line.

($P>0.087$ in all three cases). Instead, we uncovered an association between the *rpsL* background and the effects of second-step mutations, particularly with the 43N and 91Q mutations associating with aggravating and mitigating effects, respectively ($P<0.038$ in both cases, chi-squared test).

### (e) Second-step mutations mostly target the ribosome and electron transport chain

To determine the genetic underpinnings of multi-aminoglycoside resistance, we performed whole-genome sequencing on one isolate from each of the 14 combinations of genotype and treatment that produced second-step mutants (Fig 1C and 1D). Additionally, we expanded this collection with four extra independent isolates randomly selected within each treatment group. We observed a total of 26 mutational events of different classes across the 18 strain collection (Table 1). The most prevalent alterations were non-synonymous single-nucleotide polymorphisms (SNPs), constituting 50% of the total mutations. Among these, GC→AT transitions alone accounted for 46.2% of all base substitutions, consistent with previous reports indicating their predominance in bacterial mutation biases [83]. We also identified a substantial fraction of mutations caused by the mobilization of Insertion Sequences (IS), comprising 38.5% of the total. Specifically, we observed five insertions caused by IS1, four insertions caused by IS5, and a single large deletion spanning 127 loci mediated by IS5. The ancestral genome contains nine copies of IS1 and six copies of IS5. While these numbers can vary among strains and species, these ISs commonly underpin adaptive mutations in *E. coli* and related bacteria, either by altering gene expression or directly disrupting the functionality of genes relevant for adaptation [84,85]. Instances of small indels, involving insertions or deletions of a few bases, were relatively rare, with only three instances among the 26 observed mutations.

We observed that mutations mapped to either the coding or the regulatory region of 14 different genes. What evidence do we have for the adaptive significance of these mutations? Repeated alterations of the same genetic elements across independent populations are generally considered a signature of positive selection (*i.e.*, parallel evolution) [86]. A glance at Table 1 indicates that such parallel evolution was common among our strains. For instance, genes *fusA* and *cydA* were each repeatedly hit in five or more independent strains. As expected, parallelism was most

Table 1. **All mutations detected among the second-step mutants.**

| back. | drug | ribosome | | | | electron transport chain | | | | | | other | | | | | rel. fitness |
| | | rps | | | fusA | cyd | | ubi | | | atpG | | | | | | |
| | | D | E | L | | A | C | E | F | H | | cadC | sbmA | yagU | xdhA | pflD | |
| 43N | KAN | | | | V415L | | | | | | | | | | | R575S | 0.61 |
| 43T | | | | | | | A64E | | | | | | | | | | 0.92 |
| 88E | | | | | | IS5 (+ 4) | | | | | | | | | | | 0.59 |
| 88R | | | | | R671S | | | | | | | | | | | | 1.06 |
| 91Q | | | | | A678V | | | | | | | | | | SNP (-270) | | 0.57 |
| 91Q | | | | E62K | | | | | IS5 (+196) | | | | | | | | 0.76 |
| 43N | TOB | | | | | | | | | +1 bp | | | IS1 (-148) | | | R575S | 0.53 |
| 43N | | | | | P610L | | | | | | | | | | | R575S | 0.96 |
| 43T | | | | | | | | | IS1 (+744) | | | | | IS1 (-229) | | | 1.05 |
| 91Q | | T86I | | | | | | H215P | | | | | | | | | 0.68 |
| 43N | GEN | | | | I645N | IS5 (-149) | | | | | | | | | | R575S | 0.73 |
| 43N | | | | | | IS1 (-124) | | | | | | | | | | R575S | 0.93 |
| 91Q | | | G102S | | | | | | | | Δ1 bp | | | | | | 0.52 |
| 43N | AMK | | | | | IS1 (-167) | | | | | | | | | | R575S | 0.66 |
| 43T | | | | | | IS5 (-149) | | | | | | | | | | | 0.88 |
| 43T | | | | | P610L | | | | | | | | | | | | 0.83 |
| 88R | | | | | | | | | | | | Y504S | IS5 Δ126,318 bp | | | | 0.29 |
| 91Q | | | | | | Δ7 bp | | | | | | | | | | | 0.53 |

Notes: numbers in parentheses correspond to intergenic sites, indicating the distance to the start of the corresponding gene. Short insertions and deletions are indicated with a plus (+) or delta (Δ) sign followed by the number of base pairs affected.

common at higher levels of functional organization; two functional modules, the ribosome and the electron transport chain, contained 80.8% of all detected mutations (Table 1). In contrast, parallelism at the nucleotide level was rare, with only two specific mutations found more than once (one SNP in *fusA* and one IS insertion in the upstream region of *cydA*), each occurring only twice. In addition to parallelism, further support for the likely adaptive nature of these mutations comes from our understanding of the mechanisms underlying aminoglycoside uptake and their mode of action. Among the 14 genes identified (Table 1), the link to aminoglycoside resistance is relatively clear for 10 of them. We can thus classify the mutations into three broad groups based on the presumed mechanism mediating resistance. The first group includes genes encoding either protein S12 (*rpsL*) or some of its closely interacting partners, such as protein S4 (*rpsD*), protein S5 (*rpsE*) and elongation factor G (*fusA*). Of the 26 observed mutations, 9 map to these genes, with 6 occurring in *fusA* alone. These mutations likely compensate for the physical interference caused by aminoglycosides during the translocation step of protein synthesis, and mutations in these genes have already been implicated in aminoglycoside resistance [87,88].

The second group comprises genes expected to influence aminoglycoside uptake. Aminoglycosides enter the cell in an energy-dependent manner, relying on an active electron transport chain to generate and maintain the proton-motive force [55,89]. One of the systems responsible for this electrochemical gradient is the cytochrome bd-I oxidase, which couples ubiquinol oxidation with $O_2$ reduction to move protons from the cytoplasm to the periplasm [90]. Among the 26 mutations observed, 11 are within genes involved in this process: 6 are within the *cydAC* operon, encoding the two major subunits of the cytochrome bd-I oxidase, and 5 are in genes related to ubiquinone biosynthesis (*ubiE, ubiF, ubiH*) (Table 1). These mutations possibly reduce aminoglycoside uptake by interfering with the generation of the proton-motive force, as suggested in previous reports linking mutations in the electron transport chain to aminoglycoside resistance [87,91,92]. In this line, we also observed a 1 bp deletion in *atpG*, the gene coding for the gamma subunit of the ATP synthase. Disruptive

mutations in this gene have been detected in gentamicin resistant *E. coli* strains, possibly causing proton leakage and thus dissipating the proton-motive force [87,93].

The third group includes a series of genes whose functional relevance is uncertain. One gene with a plausible connection to aminoglycoside resistance is *cadC*, which encodes a transcriptional regulator involved in the biosynthesis of the polyamine cadaverine. Polyamines have been suggested to protect cells from reactive oxygen species, which accumulate during aminoglycoside stress and are a major contributor to cell death [89,92,94]. A case more difficult to interpret is posed by the IS5-mediated, 126.3 *Kbp* deletion observed in one AMK-selected lineage. This deletion spans 127 loci, including the upstream regions of *sbmA* and *yagU*, which are the target of IS1-mediated insertions occurring in two other independent isolates (Table 1). Mutations in *sbmA*, encoding a putative ABC transporter, confer resistance to antimicrobial peptides and have also been linked to kanamycin resistance in experimentally evolved *E. coli* strains [95,96].

However, no clear link is apparent with mutations in the remaining loci. For instance, the insertion of IS1 upstream of *yagU*, encoding an inner membrane protein that contributes to acid resistance (the preceding gene, *paoA*, encodes an aldehyde dehydrogenase). A similar situation applies to mutations in the intergenic region between *ygeR* and *xdhA*, which encode a predicted regulatory protein and a subunit of a xanthine dehydrogenase, respectively. Given their co-occurrence with mutations in other genes with a clearer link to aminoglycoside resistance (*fusA, ubiH;* Table 1), these changes seen upstream of *yagU* and *xdhA* may well be interpreted as neutral, passenger mutations. Special consideration merits the final locus on this list, *pflD*, which encodes a putative formate acetyltransferase. Mutations in *pflD* appeared in all derivatives of the K43N background, suggesting it was a passenger acquired during selection of this first-step mutant (see Methods). To assess its potential impact, we examined two additional, independently isolated K43N mutants. After confirming that neither carried a mutation in *pflD* or elsewhere in the genome, we found that both displayed susceptibility profiles, fitness costs, and mutation rates indistinguishable from the original K43N strain (S1 Table and S5 Fig), supporting the interpretation that the *pflD* mutation is a neutral passenger that arose during the first-step selection.

It is worth noting that 8 out of the 18 isolates appeared to acquire two mutations during the second-step selection, despite undergoing a single overnight incubation followed by plating on selective agar. Considering the relatively high mutation rates observed for different aminoglycosides ($1 \times 10^{-6} - 10^{-8}$, Fig 1C and 1D), it seems improbable that these two mutations arose during the overnight period. This would imply extremely high mutation rates ($10^{-3} - 10^{-4}$ per gene per generation), orders of magnitude higher than the typical rates observed for adaptive mutations [97,98]. One possibility is that one of these mutations was merely a neutral passenger. However, this is unlikely to explain all 8 occurrences for two main reasons. First, while the probability of observing a mutation in a random gene across the genome is in the order of $10^{-3}$ per gene per generation [11], the adaptive mutation would still need to fall within the same range to account for the observed values. Second, in at least four of the mutants, both mutations are located in genes related to either the ribosome or the electron transport chain, suggesting that both mutations contribute to adaptation. Instead, our data seem consistent with a second mutation occurring and becoming predominant within the colony as it grows on the selective plate, perhaps being further favored during the re-streaking procedure used for single colony isolation. On a final note, some of the mutations without a clear adaptive role in resistance may instead represent compensatory changes, particularly in backgrounds with large initial growth impairments (e.g., *xdhA* in the P91Q background).

### (f) Variation in evolvability results from positive epistasis

To understand why some *rpsL* mutations suppress or enhance resistance evolution, we used CRISPR-mediated gene editing to introduce the most common second-step, single-point mutation (*fusA* P610L) into both the wild-type and the less evolvable first-step background (*rpsL* K88E). We then compared its effects in these dead-end backgrounds to its effect in the most evolvable background (*rpsL* K43N). Does resistance fail to evolve due to synthetic lethality, or is the beneficial effect of the second mutation simply mitigated or magnified depending on the background? S2 Table shows that the *fusA* P610L mutation is viable and enhances resistance in both the wild-type and *rpsL* K88E backgrounds. However, this improvement is insufficient for survival under our experimental conditions. In contrast, in an *rpsL* K43N background, *fusA*

P610L alone increases resistance well beyond the selective threshold, illustrating a striking example of positive (i.e., synergistic) epistasis enhancing evolvability.

## 3. Discussion

In this study, we investigated the extent to which minor genetic differences can translate into sizable differences in evolvability. We also explored whether these putative differences conform to the general trend of declining adaptability observed in various microbial systems [30]. To this end, we set up a model system involving single-point mutations in the ribosomal gene *rpsL*, which confer resistance to the aminoglycoside streptomycin; and examined whether these mutations could serve as stepping-stones for adaptation to other classes of aminoglycosides. We confirmed they do and that they show large evolvability differences, both in the ease with which new resistant mutants arise (Fig 1) and in the magnitude of resistance increase (Fig 3). Of note, these differences are only partially explained by initial susceptibility. This is because some genotypes were incapable of sustaining further adaptation under our experimental conditions, representing evolutionary dead-ends rather than stepping-stones. However, excluding these evolvability suppressors, initial susceptibility explains 19.4% of the variation in mutation rate and 25% of the variation in resistance increase magnitude. Indeed, focusing on the subset that did produce second-step mutants, we confirmed that more susceptible backgrounds are more evolvable than less susceptible ones, thus conforming to the rule of declining adaptability.

We have documented a case study in which idiosyncratic effects stand out against a backdrop of predictable trends, bearing implications to ongoing debate on general patterns of epistasis in microbes [38,40,43–45]. Rather than providing support for a single explanation for the rule of declining adaptability, they suggest a scenario in which strong, unpredictable epistatic effects represent a sizable but minor fraction of all genetic interactions, while a majority display smoother, diminishing returns effects. The implication of this mixed view is that idiosyncratic effects, even if common, may not need to be proposed as pervasive to be consistent with a satisfactory explanation for the rule of declining adaptability. On a speculative side note, idiosyncratic effects may become dominant above a certain threshold of genetic divergence. This could make it challenging to detect statistically significant patterns in low-replication studies when strains vary at multiple loci, which may help explain the contrasting support for the rule of declining adaptability observed in previous studies on antibiotic resistance evolvability [51,53].

Yet, while idiosyncratic effects can be rationalized as emerging from the intricate peculiarities of biological systems, the question of where the diminishing returns effects emerge from remains. Do they result from redundancies at a global fitness level, or conversely within one or a few functional modules? We observed that second-step mutants predominantly acquire mutations in two cellular components: the ribosome and the electron transport chain. While these can be seen as independent modules, the predicted effect of the mutations ultimately converge on restoring efficient protein synthesis in the presence of drugs, supporting a modular epistasis explanation for the observed diminishing returns patterns. This interpretation may be amenable to rigorous testing following up on recent work modeling the growth-dependent fitness effects of streptomycin resistance mutations affecting drug uptake and ribosome binding [91].

Beyond the debate on general patterns of epistasis, the existence of drastic evolvability differences among genotypes differing just by single-point mutations merits further consideration. Previous studies identifying evolvability suppressors (or potentiators, the reverse side of the same phenomenon) have involved the absence or presence of entire genes, typically encoding elements with clear functional relevance to antibiotic resistance, such as efflux pumps, transcriptional regulators, and membrane transporters [26,48,50,52,53]. Our findings are noteworthy in this context, as they reveal that substantial disparities in evolvability can arise from single-point variants within a single gene. If these within-gene evolvability suppressors are common to other model systems across bacteria, they would substantially expand the range of potential targets for strategies aimed at harnessing unwanted evolution, potentially aiding in efforts to promote evolutionary robustness in biotechnology [47] and combat multidrug resistance in clinical microbiology [49].

It is intriguing that, in our system, most cases of evolutionary dead-ends involved either arginine (R) or glutamic acid (E) mutations at position K88. For the latter, reconstruction experiments reveal a strong negative epistasis between mutations

in *rpsL* and *fusA;* but what might be the underlying mechanisms causing mutations at this site to behave this way? A look at high-resolution crystal structures reveals nothing particularly special that may distinguish residue K88 from residues P91 and K42. Indeed, residues K88 and P91 lie within the same conserved loop of S12, and both are in close proximity to K42 in three-dimensional space (S1A Fig). And yet, this is not the first report of idiosyncratic behavior of mutations at this site. Previous studies revealed that K88R mutants survive better than K43N inside macrophages, both in *E. coli* [99] and *Legionella pneumophila* [100]. This increased survival has been linked to either increased avoidance of macrophages' immune proteins or to reduced rates of protein oxidative damage, but a precise molecular mechanism remains elusive. The other variant, K88E, has been shown to sustain higher translation rates than K42N and K42T mutants under the starvation stress of late growth phases [101]. Of note, protein S12 undergoes post-translational modification at position D89, which sits adjacent to K88. One possibility is that the idiosyncratic behavior of K88 may be linked to alterations in this post-translational modification, still under scrutiny but presumed to be either structurally or functionally important [102].

In any case, whatever peculiar effects mutations in K88 may have on protein S12 functioning, it still remains to be explained how these mutations may reduce evolvability without affecting the initial susceptibility to aminoglycosides. Strong intramolecular interactions, which can make combining two mutations largely unpredictable, may provide an explanation: second-site ribosomal mutations may be accessible for P91 and K42 but not to K88 mutants. However, the observation that TOB resistance can be acquired via mutations outside the ribosome argues against this possibility. An alternative explanation is that differences in susceptibility do exist, but MIC assays, due to their discrete and relatively noisy nature, may lack sufficient resolution to detect them. While plausible, we note that second-step mutants vary considerably in their susceptibilities, so it would still be remarkable that undetectable differences in MIC can lead to several-fold differences after a single selection step. A related, but perhaps more compelling possibility, is that the different mutations may vary in how they affect the binding affinities of the DOS-aminoglycosides, but that these differences are obscured in single-step mutants because a high concentration of drug molecules saturates the binding sites. By contrast, secondary mutations reducing drug concentrations below the saturation regime can reveal the differences in binding affinity, leading to drastic variations in susceptibility and, ultimately, survival among the second-step mutants.

On a final note, the fact that common single-point, streptomycin-resistance mutations in *rpsL* can serve as stepping-stones towards multiple aminoglycoside resistance is concerning. Streptomycin is still used in the developed world as a second-line antibiotic for tuberculosis treatment, and as a first-line option in many developing countries [66]. Consequently, it is unsurprising that *rpsL* streptomycin-resistance mutations are found at high-frequency among *M. tuberculosis* isolates from around the world [103,104]. Moreover, streptomycin is used extensively in agriculture worldwide for treating bacterial phytopathogen infections, with annual usage estimated in the tens of tons in the US alone [105,106]. Hence, there is ample opportunity for environmental, opportunistic pathogens to acquire *rpsL* streptomycin-resistance mutations; and human pathogens of environmental origin are known to reach clinical settings through a variety of channels [107,108]. It is a worrying possibility that a fraction of circulating pathogenic strains may already be primed to develop resistance to DOS-aminoglycosides, especially if this resistance can occur without any cost or may even confer a fitness advantage in the absence of drugs (Fig 3). AMK and KAN are important tools in the treatment of Multi-Drug-Resistant (MDR) tuberculosis, and together with GEN and TOB are used routinely against Gram-negative bacteria [109,110]. In this light, our results underscore the importance of further work identifying stepping-stone mutations as potential targets for surveillance, leading to an improved antibiotic stewardship that may preserve the efficacy of our current arsenal of antibiotics.

## 4. Materials and methods

### (a) Strains and culture conditions

All strains are derivative of *Escherichia coli* K-12 strain MG1655, kindly provided by Dr. A. Rodríguez-Rojas (University of Veterinary Medicine, Vienna, Austria). Culture medium was Lysogeny broth (LB, also known as Luria broth; Miller's modification: 10g/L sodium chloride, 10g/L tryptone and 5g/L yeast extract) or LB agar plates, which were purchased in powder

from Condalab (Spain). All cultures were incubated at 37°C for 24 h. Liquid cultures were propagated in 15 mL vials with 3 ml of medium and shaken at 180 rpm.

## (b) Selection of first-step mutants

We used Luria-Delbrück fluctuation assays to isolate independent streptomycin-resistant mutants. Briefly, 100 μL of a 1:10$^4$ dilution of a saturated overnight culture was used to inoculate a fresh batch of multiple independent liquid cultures. This dilution step, standard in fluctuation experiments, minimizes the likelihood of carrying over preexisting resistant mutants from the initial overnight culture: given a streptomycin-resistance frequency of ~$10^{-9}$, the expected number of resistant cells in the 100 μL inoculum is ~$10^{-5}$, giving a probability of ~ 0.001% that any resistant cell is transferred to a fresh culture. After overnight incubation, 100 uL aliquots of these cultures were plated onto LB agar supplemented with streptomycin (100 mg/L final concentration). Given the low mutation rate and high fitness cost associated with spontaneous streptomycin resistance in *E. coli*, we deemed it optimal to screen up to three independent colonies per plate to discover sufficient mutant diversity while minimizing workload [111]. These mutations were detected by standard PCR and Sanger sequencing techniques using primers 5'-GTC AGA CTT ACG GTT AAG CAC C-3', 5'-GAA CTT CGG ATC CGG CAG AAT T-3'. Sanger sequencing services were provided by Macrogen Inc. (South Korea). Overnight cultures of the verified mutants were stored with 15% glycerol in a -80 °C freezer. Streptomycin sulfate was purchased from Thermo Fisher (Spain, CAS #3810-74-0).

At this stage, we assumed that strains had only acquired a *rpsL* mutation, although we could not rule out the rare possibility of passenger mutations occurring elsewhere in the genome. With a genome-wide mutation rate of ~$10^{-3}$ [112], the expected frequency of co-occurring passenger and resistance mutations is ~$10^{-12}$, making the chance of selecting a double mutant in a 10⁹-cell population roughly 0.1%. While low, this estimate reflects an average across many cultures and does not account for early-arising mutations (i.e., "jackpot" events), which could raise the probability several-fold in a given population [111]. Later on, whole-genome sequencing revealed that the first-step K43N mutant carried a non-synonymous mutation (R575S) in the *pflD* gene, encoding for a putative formate acetyltransferase. As explained in the Results section, two additional K43N isolates—confirmed by whole-genome sequencing to lack mutations in *pflD* or elsewhere—showed phenotypes indistinguishable from the original strain (S1 Table and S5 Fig), supporting the interpretation that *pflD* is a neutral passenger. In retrospect, since multiple generations of growth are inescapable to generate the resistance mutations of interest, the only safeguard against rare co-occurring passenger mutations would have been whole-genome sequencing of first-step mutants prior to second-step selection—at the cost of a slower turnaround. Anyway, aside from *pflD*, no secondary mutations were consistently present across derivatives of any other background, demonstrating that no additional passengers were acquired during selection of first-step mutants (Table 1 shows all detected mutations across the second-step derivatives).

## (c) Selection of second-step mutants

We used a similar protocol to the one described for first-step mutants, with three appropriate modifications. First, after the initial dilution of a saturated overnight culture, we used 5 parallel liquid cultures per selection treatment. Second, 100 uL aliquots were plated onto selective solid medium supplemented with amikacin (50 mg/mL), gentamicin (50 mg/mL), kanamycin (50 mg/mL) and tobramycin (16 mg/mL). After re-streaking under the same conditions to ensure colony purity, overnight cultures of the verified mutants were stored at -80 °C with 15% glycerol. Third, $10^{-6}$ dilutions from the overnight parallel cultures were plated onto LB agar plates without antibiotics to determine the total number of viable cells. Amikacin disulfate salt was purchased from Thermo Fisher (Spain, CAS #39831-55-5), gentamicin sulfate was purchased from Thermo Fisher (Spain, CAS #1405-41-0), kanamycin Sulfate was purchased from Thermo Fisher (Spain, CAS #25389-94-0) and tobramycin was purchased from Thermo Fisher (Spain, CAS #32986-56-4).

### (d) Mutation rate estimation

Mutation rates and their 95% confidence intervals were estimated from resistant mutant counts and viable cell counts obtained from the Luria-Delbrück fluctuation assays described above, using the functions *newton.LD.plating()* and *confint. LD.plating()* from the R package rSalvador [113]. Comparisons of mutation rates among independent fluctuation experiments were performed using likelihood ratio tests (function *LRT.LD.plating()*), explicitly accounting for differences in final population sizes and partial plating, two important confounding factors.

### (e) Growth rate assays

We measured the maximum growth rate in the absence of drugs for the wild-type and for each selected first-step and second-step mutants. While maximum growth rates are just a component of fitness, in nutrient-rich media they are generally a predominant one and correlate well with competitive fitness [114]. Strains retrieved from the -80ºC freezer were used to inoculate liquid overnight cultures. On the next day, cultures were standardized to an optical density of 0.01 (600 nm) using the Ultrospec 10 cell density meter (Harvard Bioscience, UK), and 33 µl aliquots were inoculated into a final volume of 200 µl in 100-well Honeycomb 2 multi-well plates in triplicate. These multi-well plates were then incubated at 37°C for 48 h with constant shaking with settings amplitude "medium" and speed "normal", and optical density was monitored at 10 minutes intervals using a Bioscreen C Pro microplate absorbance reader (Oy Growth Curves Ab, Finland). Maximum slope analyses were done with a custom script made in R language to perform linear regression over 1 hour sliding windows.

### (f) Antimicrobial susceptibility assays

We determined the minimum inhibitory concentration (MIC) for the wild type and all selected first-step and second-step mutants using standard broth microdilution methods in accordance with the recommendations of the Clinical and Laboratory Standards Institute. Strains retrieved from the -80ºC freezer were used to inoculate liquid overnight cultures. On the next day, these cultures were centrifuged and standardized to a 0.5 McFarland, which marks approximately $1 \times 10^8$ CFU/ml in *E. coli* [115]. Cultures were then diluted at a ratio of 1:150, and 100 µl aliquots were inoculated into a final volume of 200 µl into microtiter plates filled with appropriate concentration gradients of the different compounds. The MIC was defined as the lowest concentration that completely inhibits bacterial growth after 24 hours of static incubation at 37ºC. Results are reported as the median of three replicates. Ciprofloxacin (CAS #85721-33-1) and tetracycline hydrochloride (CAS #64-75-5) were purchased from Merck (Spain). Chloramphenicol (CAS #56-75-7), neomycin sulfate (CAS #1405-10-3), spectinomycin sulfate (CAS #22189-32-8), and triclosan (CAS #3380-34-5) were purchased from Thermo Fisher Scientific (Spain). Chloramphenicol and triclosan were dissolved in ethanol, tetracycline was dissolved in 0.1 N HCl, and all others were dissolved in water.

### (g) CRISPR/Cas9-mediated gene editing

We used CRISPR/Cas9 editing to introduce the most common second-step mutation (*fusA* P610L) into both the wild-type and *rpsL* K88E backgrounds, following a protocol described elsewhere [116]. A synonymous PAM-inactivating mutation (SPM) was incorporated into the donor DNA to enhance efficiency [117]. For donor DNA construction, a ~1 kb fragment surrounding *fusA* P610L from a second-step mutant was amplified from a second-step mutant using Phusion DNA polymerase (Thermo Scientific, USA) and primers fusA_500_Fw and fusA_500_Rev (S3 Table). The PCR product was cloned into pTOPO-Blunt II (Thermo Scientific, USA), and the SPM was introduced via site-directed mutagenesis using primers fusA_SDM_Fw and fusA_SDM_Rev (S3 Table). The product was digested with Fastdigest DpnI (Thermo Scientific, USA) for 2 hours, transformed into chemically competent *E. coli* MG1655, and plated on LB agar with kanamycin (50 µg/mL). Colonies were screened by PCR and Sanger sequencing. To construct pCRISPR-gDNA, primers fusA_gDNA_Fw

and fusA_gDNA_Rev (S3 Table) were annealed to generate the gDNA. Both gDNA and pCRISPR-SacB were then digested with Fastdigest SacB (Thermo Scientific, USA), ligated using T4 DNA ligase (Thermo Scientific, USA), and transformed into *E. coli* MG1655. Transformants were plated on LB with kanamycin (50 µg/mL) and screened by PCR using fusA_gDNA_Fw and pCRISPR_SC_Rev. CRISPR/Cas9 editing was performed as previously described [113], and transformants were selected on LB with kanamycin (50 µg/mL) and chloramphenicol (25 µg/mL). Colony screening was conducted via amplification-refractory mutation screening [118] using primers fusA_SC_P610_Fw with fusA_SC_P610_wt or fusA_SC_P610_mut (S3 Table).

### (h)  Whole genome sequencing analyses

Genomic DNA was extracted using the DNeasy UltraClean Microbial Kit (Qiagen, Germany) and quantified using the Qubit 4 Fluorometer (Invitrogen, US), in both cases following manufacturer instructions. Sequencing services were provided by Novogene (UK), and performed using Illumina Novaseq 6000 with 150 bp paired-end reads. Whole genome sequencing alignment and analysis was performed using the computational pipeline breseq [119]. After filtering, the coverage depth ranged from a minimum of 178x to a maximum of 641x, with a median of 421x across all samples. Mutations were found relative to the *E. coli* reference retrieved from the National Center for Biotechnology Information (NCBI, accession number U00096.3). As is typical, we identified five differences between our parental strain and the GenBank reference, all of which have been reported in other MG1655 derivatives [120,121]. Two map to pseudogenes, two to intergenic regions, and one corresponds to the deletion of a mobile element (S4 Table). None have a clear functional connection to streptomycin resistance, and the variant closest to the *rpsL* locus is located over 0.5 Mbp away—a considerable genomic distance (~10% of the genome). Sequences are deposited in the NCBI Sequence Read Archive (PRJNA1235639).

### (i)  Statistical analyses and data visualization

All statistical tests were performed by using either built-in functions or available packages from the R programming language (version 4.2.3) through RStudio interface (version 2023.12.0+369). Pearson correlation coefficients, linear regression, pairwise Wilcoxon signed-rank tests, pairwise t-tests, Benjamini-Hochberg corrections and chi-squared tests were performed using the built-in functions *cor(), lm(), pairwise.wilcox.test(), pairwise.t.test(), p.adjust()* and *chisq.test()*, respectively. Fisher's Exact Test was performed using the function *fisher.test()* from the "stats" library. Fig 2 was generated using the "pheatmap" library. S1 Fig was generated using Pymol 2.3.0 (Schrödinger LLC, US) with the Protein Data bank entry 7OE1.

## Supporting information

**S1 Fig.  Mutations in *rpsL* identified in this work.** (A) Upper left: Crystal structure of the 30S ribosomal subunit from *E. coli* (PDB entry 7OE1), highlighting the location of ribosomal protein S12 (green) and 16S rRNA (purple). The zoomed-in view shows the A-site, where tRNA anticodons are matched with appropriate mRNA codons during translation. Residues harboring first-step streptomycin mutations, mapping to the interface between protein S12 and 16S rRNA, are highlighted in orange. A residue in which a second-step mutation was identified during selection with TOB is highlighted in cyan. (B) Abundance of different mutations in *rpsL* observed across 40 streptomycin-resistant colonies. Grey bars indicate mutations with a streptomycin-dependent phenotype, requiring sustained antibiotic binding for growth. Streptomycin-independent mutations, the focus of this work, are shown in black. Note that while differences in abundance should reflect differences in the underlying mutation rate, a one-to-one correspondence is not warranted. This is because isolates are not fully independent: up to three colonies per plate were screened to strike a balance between maximizing diversity and minimizing workload, a strategy justified by the anticipated low mutation rates and high fitness costs (Methods).
(TIF)

**S2 Fig. Mutant prevention window for the less evolvable first-step mutant.** Mutation rate for 2-DOS aminoglycoside resistance in the K88E strain (triangles, darker colors) compared with the ancestral strain (circles, lighter colors) across a gradient of lethal concentrations of amikacin (AMK, green), gentamicin (GEN, orange), kanamycin (KAN, blue), and tobramycin (TOB, pink). Antibiotic susceptibilities are shown as fold increases relative to the Minimal Inhibitory Concentration (MIC) of the ancestral strain. Values represent the mutation rate±95% confidence interval from three replicates. (TIF)

**S3 Fig. Mutation rates for other antibiotic resistance markers in first-step mutants.** Rifampicin (RIF) and fosfomycin (FOS) resistance serve as proxies for genome-wide elevations in mutation rate, as RIF-resistant mutants typically arise from mutations in *rpoB*, located approximately 759 kbp from *rpsL* in the *E. coli* genome, while FOS-resistant mutants map to *glpT* and *uhpT*, approximately 1104 kbp and 404 kbp away, respectively. Values represent the mutation rate±95% confidence interval, estimated from five parallel replicate cultures. The small differences in mutation rates are not significant (pairwise likelihood ratio tests, Benjamini-Hochberg corrected). (TIF)

**S4 Fig. Growth curves in the absence of antibiotics for all strains studied in this work.** Each line represents the average optical density over time from three parallel assays per strain. Thick curves represent the ancestor (black) and the first-step mutants (43N, green; 43T, magenta; 88R, purple; 88E, orange; 91Q, blue). Thin curves correspond to the second-step mutants, following the color coding of their corresponding first-step mutant. Maximum growth rates were estimated using a custom script that calculates the maximum slope of the natural logarithm of optical densities versus time. The script analyzed the data using a sliding window of 1 h (six points, 10 minutes apart). Within each window, the script smoothed the data using a three-step moving average and then estimated the slope by fitting a linear regression model. (TIF)

**S5 Fig. Profiling of the *pflD* mutation suggests it is a neutral passenger.** (A) Mutation rates for 2-DOS aminoglycoside resistance in the original K43N first-step mutant carrying the *pflD* mutation and two independently isolated clones confirmed to carry only the intended K43N *rpsL* mutation. Values represent the mutation rate±95% confidence interval, estimated from five parallel replicate cultures. Selection was performed at the wild-type's Mutant Prevention Concentration (MPC), as in Fig 1. The small differences in mutation rates are not significant (pairwise likelihood ratio tests, Benjamini-Hochberg corrected). (B) Growth curves in the absence of antibiotics for the original K43N first-step mutant carrying the *pflD* mutation and the two clones confirmed to carry only the K43N mutation in *rpsL*. Details as in S4 Fig. (TIF)

**S1 Table. Effect on antibiotic susceptibility conferred by *pflD* R575S in the K43N background.** (DOCX)

**S2 Table. Effects on antibiotic susceptibility conferred by *fusA* P610L across three backgrounds.** (DOCX)

**S3 Table. Primers used for CRISPR/Cas9-mediated gene editing.** (DOCX)

**S4 Table. Differences between the parental strain and the GenBank reference (U00096.3).** (DOCX)

## Acknowledgments

We thank J. Barber and L. López-Merino for proofreading the manuscript.

## Author contributions

**Conceptualization:** Laura Sánchez-Maroto, Guillem August Devin, Alejandro Couce.

**Formal analysis:** Laura Sánchez-Maroto, Guillem August Devin, Alejandro Couce.

**Funding acquisition:** Alejandro Couce.

**Investigation:** Laura Sánchez-Maroto, Guillem August Devin, Pablo Gella.

**Supervision:** Alejandro Couce.

**Visualization:** Laura Sánchez-Maroto, Pablo Gella, Alejandro Couce.

**Writing – original draft:** Alejandro Couce.

**Writing – review & editing:** Laura Sánchez-Maroto, Guillem August Devin, Pablo Gella, Alejandro Couce.

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
