## [Decision Letter · Decision Letter 0]

19 Jul 2024

Dear Dr Couce,

Thank you very much for submitting your Research Article entitled 'Idiosyncratic evolvability among single-point ribosomal mutants towards multi-aminoglycoside resistance' to PLOS Genetics.

The manuscript was fully evaluated at the editorial level and by independent peer reviewers. The reviewers appreciated the attention to an important problem, but raised some substantial concerns about the current manuscript. Based on the reviews, we will not be able to accept this version of the manuscript, but we would be willing to review a much-revised version. We cannot, of course, promise publication at that time.

If you decide to revise the manuscript for further consideration at PLOS Genetics, please aim to resubmit within the next 60 days, unless it will take extra time to address the concerns of the reviewers, in which case we would appreciate an expected resubmission date by email to plosgenetics@plos.org.

If present, accompanying reviewer attachments are included with this email; please notify the journal office if any appear to be missing. They will also be available for download from the link below. You can use this link to log into the system when you are ready to submit a revised version, having first consulted our Submission Checklist .

PLOS has incorporated Similarity Check , powered by iThenticate, into its journal-wide submission system in order to screen submitted content for originality before publication. Each PLOS journal undertakes screening on a proportion of submitted articles. You will be contacted if needed following the screening process.

To resubmit, log into your Editorial Manager account and select the option 'Revise Submission' in the 'Submissions Needing Revision' folder.

We are sorry that we cannot be more positive about your manuscript at this stage. Please do not hesitate to contact us if you have any concerns or questions.

Yours sincerely,

Isabel Gordo

Guest Editor

PLOS Genetics

Sean Crosson

Section Editor

PLOS Genetics

All reviewers considered that the manuscript addresses an important problem in antibiotic resistance and does so in the context of a more global process of declining adaptability. The work has the potential to bring novel and interesting results to the broad readership of PLoS Genetics.

However, we concur with issues raised by the reviewers regarding the need for further experiments to better support the conclusions of the manuscript.

In particular, addressing the issue of off-target mutations and estimation of mutation rate are important issues that need to be addressed.

Furthermore, attention needs to be given to the supplementary data (which needs to be improved and provide files with the data plotted in figures) and to data accessibility (which requires uploading sequencing data to a repository so that it becomes open access).

Reviewer's Responses to Questions

**Comments to the Authors:**

Reviewer #1: The authors investigated the effect of first-step streptomycin resistance mutations in rpsL on the ability to evolve resistance to 2-DOS aminoglycosides. They observed that in most, but not all cases, first-step rpsL mutants have second-step mutations that are not accessible from the wild type at the selection concentration. The increase in resistance is lower for first-step mutants with higher initial resistance, consistent with a trend of declining adaptability. The authors also found substantial cross-resistance between mutants selected for 2-DOS aminoglycosides, while streptomycin resistance is largely retained. Growth in the absence of antibiotics is reduced in 4 out of 5 first-step mutants, a fitness cost that may be exacerbated or reduced in the second selection step. WGS revealed that second-step resistance mutations also target ribosomal proteins and components of the electron transport chain.

How existing genetic diversity affects the evolvability of resistance to antibiotics is an interesting and highly relevant question, important for the development of new strategies against resistance evolution. This work shows that readily accessible single-step streptomycin resistance mutations can facilitate adaptation to other aminoglycosides, which is potentially clinically relevant. The results are clearly presented and the experimental design adequately addresses the question posed. The conclusions are mostly well supported by the experimental evidence, with one major exception.

Major comments:

1. Several first-step streptomycin mutants have been described as evolutionary dead ends that prevent the evolution of resistance to 2-DOS aminoglycosides. However, the experimental design only shows that several first-step mutants do not increase the evolvability over the wild type. Mutant frequencies are concentration dependent in this assay and the choice of concentration therefore plays a key role. Here, the chosen mutant prevention concentration (MPC) of the wild-type, although justified to avoid mutants without the initial rpsL mutations, does not allow a more fine-grained comparison between the evolvability of the wild type and the first-step mutants in those cases where no mutants are observed at the MPC.

2. Similarly, the claim that the beneficial mutation rate is increased is doubtful, as most of the results can be explained by a reduced susceptibility to the respective aminoglycoside alone, allowing second-step mutations to overcome the resistance level with the same resistance increase that would be insufficient in the wild type.

Minor comments:

1. Inconsistency in description of MIC change: Figure 1E and Figure 2 show different values for MIC fold increase. Presumably, Fig. 2 shows resistance values on a log2 scale. This needs to be clarified and would be easier to follow if it were consistent across different figures.

2. Fix the inconsistency of the mutant order in the plot for 88R and 88E (compare e.g. Fig. 1 and 2).

3. Lines 247-250: The reduction in streptomycin resistance in second-step mutants described for K88R is not visible in Figure 2, possibly because the color scale is saturated. The reduction for P91Q is drastic and visible, but only for KAN and TOB and not for all conditions.

4. The statistical description of correlations is somewhat unusual: the correlation is quantified by Pearson's R, while the p-value comes from a linear regression (e.g., in line 274 referring to Figure 3A). It would be more conventional to report the Pearson's correlation and the corresponding p-value and/or the slope of the regression and its significance.

5. Line 260: (4/42, <1%) should read (4/42, <10%).

6. It is difficult to keep track of specific first-step mutants and individual replicates. Figure 1E and Figure 3 could be improved if the first-step mutants could be identified, e.g. by different symbols in the scatter plot. Also, in Table 1, the replicates cannot be traced to compare the fitness costs of the replicates in Fig. 4 with the observed mutations.

7. The determination of mutant frequencies is not described in the Methods section.

8. Since the fitness cost of resistance increase was measured, is there a correlation between null fitness and MIC of replicates?

Reviewer #2: Attached.

Reviewer #3: The manuscript entitled "Idiosyncratic evolvability among single-point ribosomal mutants towards multi-aminoglycoside resistance", by Sánchez-Maroto et al., aims to determine whether the presence of streptomycin resistance mutations can favor or hamper appearance of additional mutations that confer resistance to other aminoglycosides. The main conclusions of the authors are that (i) most streptomycin resistance mutations studied facilitated evolution of resistance to other aminoglycosides, (ii) in a few cases, this did not happen, (iii) more fit genotypes showed higher rate of acquisition of beneficial mutation, but smaller effect sizes than less fit genotypes, (iv) second-step resistance mutations mostly confer cross-resistance across aminoglycosides, and (v) in some cases, second-step resistance mutations can increase or decrease the fitness cost caused by first-step resistance mutations.

The manuscript clear, well written and easy to follow, being therefore accessible to the broad readership of PLOS Genetics. The observations described in the manuscript are timely and relevant, as the observed increased evolvability of antibiotic resistant strains with respect to susceptible clones represents an additional challenge in the already complex antibiotic resistance crisis. However, I have a number of concerns that, if addressed, would ratify that the conclusions reached by the authors are properly supported by the data provided. These concerns are discussed below, divided as major and minor, and organized by relevance. Line numbers refer to those in the pdf version of the manuscript.

MAJOR CONCERNS

1. The fact that most first-step streptomycin resistant mutants can evolve resistance to aminoglycosides at concentrations that their susceptible ancestor can not is the core observation of the study. A possible explanation for that observation is that overall mutation rate of streptomycin resistant mutants is higher than that of the susceptible ancestor. In order to rule out this possibility, authors estimated mutation rates by calculating the frequency of appearance of rifampicin resistant mutants in all genotypes (Figure S2). I see two problems with this experiment:

i) Strong epistatic interactions between rifampicin and streptomycin resistance mutations have been repeatedly observed, which causes some streptomycin and rifampicin resistant mutants to show large fitness costs (PMID: 1094452; PMID: 322146; PMID: 19629166; PMID: 23206139; PMID: 26130082; PMID: 28419091; PMID: 33830249). Thus, in certain streptomycin resistant backgrounds, spontaneous appearance of rifampicin resistant mutants in cultures can show a large fitness disadvantage, which influences their frequency.

ii) Based on the legend of the axis in Figure S2, I assume (this experiment is not described in Materials and Methods) the experiment measured frequency of appearance of resistant mutants instead of mutation rates (in which the aforementioned frequency is normalized to the number of generations in each genotype). This is problematic, considering that growth rates of streptomycin resistant genotypes largely differ from that of the ancestral susceptible strain (Figure S3 and Figure 4).

Because of the two reasons above, and considering the importance of measuring mutation rates for the key discoveries described in the manuscript, I think mutation rates should be properly estimated by selecting for mutants that show no epistatic interactions with streptomycin resistance mutations (e. g. nitrofurantoin resistance mutations), and normalizing their frequencies of appearance to the number of generations that each genotype underwent during the experiment. Proper estimation of mutation rates of genotypes carrying the K43T, K43N and K88E substitutions is particularly important, as these three genotypes have been demonstrated to show increased SOS induction (PMID: 33830249), which typically increases mutation rate.

2. I find surprising that the streptomycin resistant alleles selected in the firs-step selection were not transferred to a clean background by P1 transduction, which is technically trivial and would dramatically decrease the chances of these genotypes to carry additional mutations (such as compensatory ones) together with the ones conferring resistance to streptomyicin. This is particularly relevant, considering that almost half of the 18 isolates that were subject to whole genome sequencing showed two mutations apart from the one conferring resistance to streptomycin. Indeed, sequencing of the strain carrying the K43N substitution in RpsL showed an additional mutation (discussed in section b of materials and methods, lines 518-525), demonstrating that mutations other than those conferring resistance to streptomycin were selected during the process of obtaining the first-step mutants. It is not clear whether or not all first-step streptomycin resistant genotypes were subject to whole genome sequencing and, if yes, what were the results of such analysis (number of genotypes showing additional mutations, number and nature of those mutations, etc.). One could argue whole genome sequencing of fist-step streptomycin resistant mutants is required in order to perform a proper comparison with those of second-step mutants; I was surprised not to find this data in the manuscript. In any case, since the conclusions of this work are based on the assumption that all observed differences between the susceptible ancestral and first-step mutants (in evolvability, fitness, etc.) are due exclusively to the presence of the streptomycin resistant alleles, it is paramount to determine if any other streptomycin resistant first-step mutant carries additional mutations, how many, which ones, etc.

3. Connected with point 2 above, selection of first-step mutants was performed following two overnight growths of susceptible bacteria. Indeed, according to section b of materials and methods (lines 508-511), seems like a unique bacterial culture was grown overnight and then multiple dilutions of the very same culture were grown overnight again, prior to plating onto LB agar supplemented with streptomycin, to select for mutants. I see two main issues connected with this experimental procedure:

i) first, if all experimental replicates come from a single overnight culture, streptomycin resistant clones appearing in that single culture will propagate to the replicates, biasing the results towards increased relative abundance of the specific streptomycin-resistant allele selected in the first growth. The same would occur with any other mutation (either compensatory or highjacker) that may arise in that initial culture. Indeed, the second growth might select for compensatory mutations to streptomycin-resistant mutants that may have appeared in the first growth.

ii) second, he fact that bacterial cells are grown during two consecutive overnight periods dramatically increases the chances of appearance and selection of mutations.

This approach is even more problematic in the case of the second-step selection, which was done in a similar manner. In that scenario, streptomycin resistant mutants were grown in two consecutive overnight periods prior to selection on plates containing different aminoglycosides. The two consecutive growth periods make very likely accumulation of mutations that compensate for the fitness cost caused by streptomycin resistant allels, especially for the genotypes with lower fitness.

4. Certain streptomycin resistant mutants fail to evolve resistance to certain antibiotics at the concentration used in this study (Figure 1C). The authors interpret this as evolutionary dead-ends. I do not think this interpretation is supported by the data presented in the manuscript, as this selection was done at concentrations were the ancestral susceptible strain is not able to develop resistance either. According to the author's interpretation, this would imply that wild-type E. coli represents a evolutionary dead end for aminoglycoside resistance mutations, which I do not think is a correct notion. In order to prove that these streptomycin resistant mutants unable to develop resistance to aminoglycosides are truly evolutionary dead-ends, the Mutant Prevention Concentration (MPC) should be determined for these genotypes (as it shown in Figure 1B). Only if MPC of these particular streptomycin resistant alleles is significantly lower than that of their susceptible ancestor, the intepretation of these genotypes as evolutionary dead ends would be substantiated.

MINOR CONCERNS

A. Fitness of the all genotypes used for virulence experiments were estimated by obtaining growth curves based on optical density at 600nm (Figure S3). Albeit fast and simple, this method is prone to artifacts, since optical density only correlates with cell density at low values (PMID: 322614) and is affected by bacterial cell size and shape (discussed, for instance, in PMID: 36228033), both of which can be altered in mutants (PMID: 28886685). In my opinion, relative fitness measurements would be more reliable if a more accurate method, such as CFUs measurement or competitive fitness assays, would be used.

B. I am surprised the allele causing the substitution K43R did not appear among the streptomycin resistant genotypes, as it is a commonly selected allele and, importantly, it does not cause any defect in fitness, which intuitively would cause it to be fairly represented in any cultura it may appear, compared to other alleles causing large fitness costs. What's the author's interpretation of this result?

C. I am similarly surprised the allele causing the substitution P91Q was not classified as streptomycin dependent, considering that has been previously described as such (PMID: 8412684). How do the authors explain this disagreement?

TEXT CORRECTIONS AND SUGGESTIONS

Lines 221-222. the text says: "Our analysis revealed a moderate positive correlation between initial susceptibility and mutant frequencies". The analysis shown in panel E of Figure 1 compares mutant frequencies with change in MIC. I suggest to correct the sentence above.

Line 223. I believe the authors mean Figure 1E (there is no panel H).

Line 225. the text reads "higher rates of beneficial mutations". I think the term "beneficial mutations" is confounding here, as resistance mutations are only beneficial in the presence of antibiotics, and when they appear (in the overnight growth prvevious to plating onto medium supplemented with antibiotics) are mostly deleterious. I would replace the term by "resistance mutations".

Line 249. Based on the information in Figure 2, it is not clear why the genotype causing the substitution K88R is mentioned in this sentence, as second-step mutations carrying it do not seem to show any difference in streptomycin resistance. Second-step mutants resistant to kanamycin that carry the substitution K88E show a slight decrease, but the ancestral K88E is also less resistant than other alleles.

Line 260: I believe the authors mean "4/42, < 10%).

Line 263. I suggest the authors to replace the word "modules" by "functions".

Line 297. I believe the authors mean "with the exception of K88R".

Line 528. I believe this section is c).

**Have all data underlying the figures and results presented in the manuscript been provided?**

Reviewer #1: **No: ** Supplemental files with the data shown in the figures would be helpful.

Reviewer #2: **No: ** MIC values should be added as a supplementary table. WGS data should be uploaded on an open repository.

Reviewer #3: Yes

PLOS authors have the option to publish the peer review history of their article (what does this mean? ). If published, this will include your full peer review and any attached files.

**Do you want your identity to be public for this peer review?** For information about this choice, including consent withdrawal, please see our Privacy Policy .

Reviewer #1: No

Reviewer #2: No

Reviewer #3: **Yes: ** Roberto Balbontín

---

## [Decision Letter · Decision Letter 1]

19 Apr 2025

PGENETICS-D-24-00553R1

Idiosyncratic evolvability among single-point ribosomal mutants towards multi-aminoglycoside resistance

PLOS Genetics

Dear Dr. Couce,

Thank you for submitting your manuscript to PLOS Genetics. After careful consideration, we feel that it has merit but does not fully meet PLOS Genetics's publication criteria as it currently stands. Therefore, we invite you to submit a revised version of the manuscript that addresses the points raised during this review process.

Please note that the reviewers appreciate the work you’ve done to improve the manuscript and agree that it has been significantly strengthened. However, both reviewers raise additional valid concerns regarding data presentation, terminology, and interpretation. These issues should be carefully addressed in a detailed, point-by-point response upon resubmission. 

Please submit your revised manuscript within 30 days May 19 2025 11:59PM. If you will need more time than this to complete your revisions, please reply to this message or contact the journal office at plosgenetics@plos.org. Please include the following items when submitting your revised manuscript:

We look forward to receiving your revised manuscript.

Kind regards,

Sean Crosson

Section Editor

PLOS Genetics

Aimée Dudley

Editor-in-Chief

PLOS Genetics

Anne Goriely

Editor-in-Chief

PLOS Genetics

**Journal Requirements:**

1) We have noticed that you have uploaded Supporting Information files, but you have not included a list of legends. Please add a full list of legends for your Supporting Information files after the references list.

2) Please ensure that the funders and grant numbers match between the Financial Disclosure field and the Funding Information tab in your submission form. Note that the funders must be provided in the same order in both places as well.

**Reviewers' comments:**

Reviewer's Responses to Questions

**Comments to the Authors:**

Reviewer #1: The authors have adequately responded to and implemented my comments and suggestions. The additional experiments improve the manuscript and strengthen the main conclusions.

Minor comments:

The use of the term "dead ends" for some first-step mutants, e.g., in the abstract, remains somewhat misleading and counterintuitive, since the tested case K88E (Figure S2) merely shows a lack of increased evolvability beyond wild type, as also described by the authors in the added paragraph (ll. 210).

Mutation rate determination (Figure 1, S3 and S5):

• To my understanding, the authors used 3 (first step) or 5 (second step) parallel cultures for plating and mutant counting. The mutation rate was then estimated once. However, the caption of Figure 1 can be interpreted as 5 separate mutation rate estimates and should be rephrased. Notably, 5 or fewer parallel cultures are very few for accurate mutation rate determination in a fluctuation assay.

• The mutation rate points look like boxplots and are also labeled as such in the caption of Figure S3. However, they seem to represent a single point with the estimated 95% confidence intervals as error bars in all cases. The appearance should be adjusted and the correct description added in all cases.

Figure 1: Subplots B, C and E all show mutation rates, but the axis limits are different. It would be easier to compare between conditions based on one plot range.

l. 308: The reference to Figure S3 should now be changed to S4 due to the addition of the mutation rate determination. Same in the caption of Figure 4 (l. 998).

Supplementary Information l. 106: The CRISPR/Cas9 primer table should be Table S3.

Reviewer #3: I appreciate the efforts made by the authors of the manuscript entitled "Idiosyncratic evolvability among single-point ribosomal mutants towards multi-aminoglycoside resistance", by Sánchez-Maroto et al. to address the concerns expressed by me and other two reviewers. The authors performed new experiments and significantly edited the text of the manuscript, and these efforts should be commended. A fraction of my concerns are completely or partially mitigated by the data and discussions in the revised version of the manuscript. Unfortunately, there are others that, in my opinion, still require further action. These concerns are discussed below, maintaining the nomenclature of my previous comment.

MAJOR CONCERN 1. In my opinion, not calculating the average number of generations in the new experiment with fosfomycin was a lost opportunity, as could have allowed the authors to experimentally estimate mutation rates, besides the mathematical inference shown in the revised version of the manuscript. However, such inference shows no significant differences in mutation rates, and data used for it was obtained in a new independent experiment selecting for alleles that in principle do not interact epistatically with streptoymcin resistant alleles. Thus, my major concern 1 can be considered as mitigated.

MAJOR CONCERNS 2 AND 3. The low probability of sampling preexisting mutants upon 10−4 dilution is based on the assumption that the frequency of newly arisen mutants in the bacterial population remains stable since the appearance of the mutation until the moment of sampling. However, if these newly arisen mutations are neutral or beneficial (for instance, if they cause positive epistasis with other polymorphisms present in the parental background, or if they appear in a genetic background that already carried a beneficial mutation), frequency of newly appeared mutants can be much higher than the estimated ~10−9 at the time of sampling, especially if they appeared early in the growth of the previous liquid culture. Indeed, the fact the the authors found the same mutation in pflD in all derivatives of the K43N background proves that the purpose of eliminating "any mutants that may have arisen in the initial overnight culture" was defeated, and that this experimental approach does not prevent selection of passenger mutations. The effect of mutations other than the alleles conferring antibiotic resistance can only be ruled out if multiple initial cultures are run in parallel, growth steps are minimized, results from these independent initial cultures are compared, and clones carrying additional mutations are excluded from the analysis. This would require essentially repeating most of the experiments that constitute the core of this work, and I do not think requesting that would be fair to the authors at this point. An alternative way to mitigate these concerns would thus be to discuss in the main text of the manuscript the limitations of the approach selected, as well as the point discussed below.

Moreover, it is not clear if other mutations, even if they did not appear in all the replicates from the same genetic background, were detected after performing genome sequencing of the 18 strains derived from first-step mutants. The presence of any specific allele in more than one replicate derived from the same streptomycin resistant background would indicate its appearance during selection of first-step mutants (unless one assumes the unlikely scenario of simultaneous appearance of the very same allele in independent cultures). I thus think the full list of mutations found in the sequencing of the 18 isolates should be available, and appearance of parallel mutations in replicates from the same streptomycin resistant background discussed in the main text of the manuscript.

MAJOR CONCERN 4. I still think the term "evolutionary dead end" is misleading. Even more so in light of the results obtained by the authors in response to th reviewer's comments (shown in the new Supplementary Figure S2), which demonstrate that even in the most extreme case, evolvability of the so-called evolutionary dead-ends is not lower than that of the wild type (actually, is higher for one antibiotic). Consideration of the notions "evolutionary stepping-stone" and "evolutionary dead-end" as absolute categories is oversimplistic. The fact that certain mutations conferring resistance to streptomycin can open additional evolutionary pathways towards resistance to other aminoglycoside can certainly be considered an evolutionary stepping-stone. But calling other mutations "evolutionary dead-end" suggests that they cause the opposite effect; that is, that they reduce evolvability when, in reality, they simply do not affect evolvability at all. In my opinion the confusion will be eliminated if the authors simply state that some mutations constitute evolutionary stepping-stones and others do not.

MINOR CONCERN A. Table 1 and Figure S2 in PMID: 33830249 indeed show that E. coli streptomycin resistant mutants carrying the K43N substitution in rpsL show no apparent difference in cell shape and frecuency of DNA breaks (and therefore SOS induction). However, Figure S1 shows a strong induction of the SOS response in the strain carrying the K88E substitution. Since activation of the SOS response causes inhibition of cell division and thus cell elongation, this mutant is expected to be strongly affected in cell shape. Thus, I still think competitive fitness of the resistant genotypes under study against a wild-type reference would be a more accurate representation of fitness than growth curves. However, since this would involve a significant amount of work at this stage of the publication process, if other reviewers and the editor accept growth curves, I'd be happy to go with the majority.

MINOR CONCERN B. Spontaneous E. coli resistant mutants carrying the strA60 allele, causing the amino acid substitution K43R, have been repeatedly isolated by multiple laboratories in the last 50 years (e. g. PMID: 5326090; PMID: 4920897; PMID: 4560854; PMID: 19629166). It is thus not an uncommon allele to appear in E. coli. According to section b of materials and methods (lines 533-540 in the revised version of the manuscript), a unique bacterial culture was grown overnight and then multiple dilutions of the very same culture were grown overnight again, prior to plating onto LB agar supplemented with streptomycin, to select for mutants. In that scenario of a single biological replicate diluted into several replicates, biases towards particular alleles appearing early in the initial overnight cultures are expected, and therefore the absence of certain substitutions, including K43R, is not that much surprising. However, in their response, the authors mention the possibility of epistatic interactions with other polymorphisms as possible factors causing the lack of appearance of this allele in their experiments. Since the genotype of the ancestral strain used (or a strain name and its corresponding reference) is not provided in the initial nor in the revised version of the manuscript, it is possible this interpretation by the authors is correct and other polymorphisms are hampering the appearance of the allele causing the K43R substitution. In order to rule out (or confirm) this hypothesis, and considering that the ancestral genetic background is paramount in this type of studies, a clear description of the strain used is necessary and, if any mutation potentially explaining the absence of the allele causing the K43R substitution (or any other allele causing streptomycin resistance) is present in ancestral strain, this must be discussed in the main text of the manuscript.

**Have all data underlying the figures and results presented in the manuscript been provided?**

Reviewer #1: Yes

Reviewer #3: **No: ** The details of the ancestral strain used in this study are not disclosed

PLOS authors have the option to publish the peer review history of their article (what does this mean? ). If published, this will include your full peer review and any attached files.

**Do you want your identity to be public for this peer review?** For information about this choice, including consent withdrawal, please see our Privacy Policy .

Reviewer #1: No

Reviewer #3: **Yes: ** Roberto Balbontín

**Figure resubmission:**
---

## [Decision Letter · Decision Letter 2]

13 Jul 2025

PGENETICS-D-24-00553R2

Idiosyncratic evolvability among single-point ribosomal mutants towards multi-aminoglycoside resistance

PLOS Genetics

Dear Dr. Couce,

Thank you for your careful attention to the reviewers’ comments and concerns. As you will see from the comments, reviewer #1 is fully satisfied with your responses and revisions. With one exception, reviewer #3 is also fully satisfied. The remaining concern (number 4) of this reviewer centers on the definition of an "evolutionary dead end", as defined in published studies. We invite you to submit a revised version of the manuscript that addresses this reviewer’s points.

Please submit your revised manuscript within 30 days Aug 12 2025 11:59PM. If you will need more time than this to complete your revisions, please reply to this message or contact the journal office at plosgenetics@plos.org. Please include the following items when submitting your revised manuscript:

* A rebuttal letter that responds to each point raised by the editor and reviewer. You should upload this letter as a separate file labeled 'Response to Reviewers'. This file does not need to include responses to formatting updates and technical items listed in the 'Journal Requirements' section below.

We look forward to receiving your revised manuscript.

Kind regards,

Sean Crosson

Section Editor

PLOS Genetics

Aimée Dudley

Editor-in-Chief

PLOS Genetics

Anne Goriely

Editor-in-Chief

PLOS Genetics

**Journal Requirements:**

1) Thank you for stating that "All data necessary to replicate the research findings is available at public data repositories (NCBI Sequence Read Archive, PRJNA1235639)." Please note that, though access restrictions are acceptable now, your entire minimal dataset will need to be made freely accessible if your manuscript is accepted for publication. This policy applies to all data except where public deposition would breach compliance with the protocol approved by your research ethics board. 

2) Please amend your detailed Financial Disclosure statement. This is published with the article. It must therefore be completed in full sentences and contain the exact wording you wish to be published.

3) Please ensure that the funders and grant numbers match between the Financial Disclosure field and the Funding Information tab in your submission form. Note that the funders must be provided in the same order in both places as well. Currently, these grants "PID2022-142857NB-I00 and 2023-5A/BIO-28940" are missing from the Funding Information tab.

**Reviewers' comments:**

Reviewer's Responses to Questions

Reviewer #1: The authors have addressed my remaining comments convincingly. I support the publication of the manuscript.

Reviewer #3: The authors of the manuscript entitled "Idiosyncratic evolvability among single-point ribosomal mutants towards multi-aminoglycoside resistance", have made a commendable effort to address my remaining concerns. Further changes in the text of the manuscript mitigate major concerns 1, 2 and 3 and minor concerns A and B. However, major concern 4 still remains.

I understand the intention of the authors when defining "evolutionary dead end", and I appreciate their efforts to clarify their particular definition. Unfortunately, I think this nomenclature is not only semantic, but conceptually incorrect. The authors seem to conceive a binary separation between "stepping-stone" and "dead end". That is, any mutation increasing evolvability can be considered a "stepping stone" mutation, and any mutation not increasing evolvability can be considered an evolutionary "dead end". This bipolar nomenclature does not allow to distinguish between mutations that do not affect evolvability and mutations that decrease evolvability; it thus bundles together mutations that are neutral to the phenotype under study (evolvability) with those negatively affecting it. In my opinion this significant biological and evolutionary difference cannot be ignored, making the nomenclature proposed by the authors misleading.

Regarding the literature used by the authors to discuss this point (PMID: 39261599, PMID: 36170241, PMID: 33634790, and PMID: 32561723), I fear does not really support their perspective. For instance, the paragraph where Ferrare and Good (PMID: 39261599) disscuss the concept of evlutionary "dead end" reads as follows: "The opposite behavior occurs when short-term fitness benefits are linked to long-term reductions in evolvability. In this case, modifiers that would be strongly disfavored on their own [ fix⁡(( , )→( ′ , ′ ))≪⁢1] will generally amplify the relative contribution of a direct fitness benefit (Fig. 3B,C). A striking example of this effect occurs in the extreme case of an evolutionary “dead end”, where further beneficial mutations are not available." Thus, Ferrare and Good define evolutionary "dead ends"as a specific case occurring when "short-term fitness benefits are linked to long-term reductions in evolvability". Similarly, Jagdmann et al. (PMID: 36170241) discuss the concept of evolutionary "dead end" in the context of a deletion in the secuence encoding the TetR repressor, which prevents selection of resistance to tygecycline mediated by gene amplification. In that scenario, strains carrying the deletion clearly suffer a reduction in evolvability compared to the strains not carrying it. Finally, Souque et al. (PMID: 33634790) use the term "evolutionary dead end" referring to the context of a particular experiment (which required evolving strains to survive to increasing concentrations of gentamicin), while discussing the effect of the tandem duplication of an antibiotic resistance gene (aadB) in an artificial three-cassette integron; the absence of tandem duplication at concentrations beyond 4X the MIC made the authors infer that it only conferred a small increase in gentamicin resistance, and did not allow to acquire additional mutations that further increase resistance. Thus, the phenomenon described by Souque el al. also involves a reduction in evolvability. Lukačišinová et al. (PMID: 32561723) literally study and genes that, when deleted, reduce evolvability of genotypes carrying these deletions with respect to the ancestral reference strain. This work thus also agrees with the necessity of a reduction in evolvability in order to define an evolutionary "dead end".

In conclusion, according to the literature proposed by the authors, genotypes unable to generate second-step mutants could only be considered "evolutionary dead-ends" if they suffered a reduction in evolvability with respect to their ancestral reference. Streptomycin-resistant mutants did not underwent any reduction in evolvability compared to wild-type bacteria. And wild-type bacteria did not suffer decreased evolvability with respect to itself. These genotypes simply did not benefit from the increase in evolvability caused by the acquisition of "stepping-stone" mutations. Thus, genotypes not increasing evolvability cannot be conssidered "evolutionary dead ends". I encourage the authors to define the phenotypes observed precisely, avoiding the use of terms that cannot be specifically adscribed to their observations, such as "evolutionary dead end".

**Have all data underlying the figures and results presented in the manuscript been provided?**

Reviewer #1: Yes

Reviewer #3: Yes

PLOS authors have the option to publish the peer review history of their article (what does this mean? ). If published, this will include your full peer review and any attached files.

**Do you want your identity to be public for this peer review?** For information about this choice, including consent withdrawal, please see our Privacy Policy .

Reviewer #1: No

Reviewer #3: **Yes: ** Roberto Balbontín

**Figure resubmission:**
---

## [Editor Report · Decision Letter 3]

5 Aug 2025

Dear Dr Couce,

We are pleased to inform you that your manuscript entitled "Idiosyncratic evolvability among single-point ribosomal mutants towards multi-aminoglycoside resistance" has been editorially accepted for publication in PLOS Genetics. Congratulations!

Yours sincerely,

Sean Crosson

Section Editor

PLOS Genetics

Aimée Dudley

Editor-in-Chief

PLOS Genetics

Anne Goriely

Editor-in-Chief

PLOS Genetics

**Data Deposition**

http://datadryad.org/submit?journalID=pgenetics&manu=PGENETICS-D-24-00553R3

**Press Queries**

---

## [Editor Report · Acceptance letter]

PGENETICS-D-24-00553R3

Idiosyncratic evolvability among single-point ribosomal mutants towards multi-aminoglycoside resistance

Dear Dr Couce,

We are pleased to inform you that your manuscript entitled "Idiosyncratic evolvability among single-point ribosomal mutants towards multi-aminoglycoside resistance" has been formally accepted for publication in PLOS Genetics! Your manuscript is now with our production department and you will be notified of the publication date in due course.

With kind regards,

Benedek Toth

PLOS Genetics

On behalf of:
